# The Direct and Indirect Costs of Breast Cancer in Poland: Estimates for 2017–2019

**DOI:** 10.3390/ijerph192416384

**Published:** 2022-12-07

**Authors:** Michal Seweryn, Tomasz Banas, Joanna Augustynska, Ola Lorenc, Justyna Kopel, Elzbieta Pluta, Tomasz Skora

**Affiliations:** 1Associate of the Institute—European Observatory of Health Inequalities, Calisia University, 62-800 Kalisz, Poland; 2EconMed Europe, 31-469 Krakow, Poland; 3Department of Radiotherapy, Maria Sklodowska-Curie National Research Institute of Oncology, Kraków Branch, 31-115 Krakow, Poland

**Keywords:** breast cancer, public payer expenditures, therapeutic program, breast-conserving surgery, mastectomy, direct costs, indirect costs, Poland

## Abstract

Background: In Poland, breast cancer (BC) is the most frequently diagnosed cancer in women and the second most common cause of death after lung cancer. This disease has important economic implications for patients, public payers, and the whole Polish economy. This study aimed to estimate the total National Health Fund (NHF) expenditures on the diagnosis and treatment of patients with breast cancer. In addition, the costs of productivity losses were also calculated. Methods: Cost estimation was prepared using a top-down approach. Direct cost calculations were based on data reported by NHF for patients with the diagnosis of breast cancer. Medical care costs included the following components: screening program, oncological package, surgical treatment, hospitalization, drug program, chemotherapy, radiotherapy, and outpatient care. Indirect costs in the form of absenteeism costs were calculated based on data from Statistics Poland (gross domestic product, number of employees) and the Social Insurance Institution database (the number of sick leave days). Results: Total expenditures for BC including direct costs and indirect costs amounted to EUR 305,371, EUR 332,998, and EUR 344,649, respectively in 2017, 2018, and 2019. Total healthcare costs in 2019 were EUR 4114 lower than in 2018, which resulted from the reduction in expenditure on the drug program (decrease of EUR 13,527), despite the observed increase in all remaining resources. From direct costs, the highest expense was spent on the drug program (nearly 50% of total direct costs), but this expense dropped significantly in 2019. For the remaining parameters, the costs increased year by year, of which the most expensive were surgical treatment (15%), radiotherapy (12%), and the screening program (10%). BC generated over EUR 120 thousand of social costs in 2019 and compared to 2017, there was an increase in productivity loss by 26%. Conclusions: Our results from 2017–2019 demonstrated that total expenditure for BC in Poland increased from year to year. Breast cancer generated almost EUR 345 thousand expenses in 2019, which translates into a significant burden on the public payer’s budget and the society in Poland.

## 1. Introduction

Breast cancer (BC) is the most frequently diagnosed cancer in women (25.3%) and the second most common cause of death (16.4%) after lung cancer in Poland [1]. The vast majority of BC occur in women and the number of cases is 100 times higher in women than in men. Besides sex, aging is one of the most important risk factors for BC because the incidence of BC is highly related to increasing age. Nearly a quarter of all BC cases are related to family history. Women, whose mothers or sisters had BC, are prone to this disease. The inherited susceptibility to BC is partially attributed to the mutations of BC-related genes such as BRCA1 and BRCA2. Reproductive factors: early menarche, late menopause, older age at first pregnancy, and low parity can increase the BC risk [2]. To diagnose BC, a medical examination, imaging, and biopsy are performed. BC treatment depends on the size, location, and the number of tumors as well as their pathological characteristics, i.e., tumor stage (grade 0-IV), subtype, and the presence of biomarkers. Treatment includes a combination of various therapeutic options: pharmacological treatment (chemotherapy, hormone therapy), surgery, and radiotherapy [3].

The actual BC incidence rate in Poland was equal to 119.1 to 100,000 people (age-standardized rate), which was less than the European average (EU-27–142.8). However, the mortality rate (to 100,000 people) was higher in Poland (41.8 vs. 34.1) [1]. Based on the World Health Organization’s prognosis of the estimated number of new cases of BC from 2020 to 2040, in 2040 would be 27,041 new cases of female BC in Poland, which is an increase of 9.7% over 20 years. In comparison with projected new cases in Europe, the growth in Poland would be higher than the European average (9.7% vs. 7.0%) [4]. In addition, trends based on data from 1980 to 2018 also showed significant increases in the incidence of BC in Poland [5,6].

Publication Preventable and treatable mortality statistics (Eurostat [7]) demonstrated that among females, the leading cause of death from treatable diseases was breast cancer. In Poland, this disease alone was responsible for every fifth death (21.2%) caused by treatable diseases/conditions among females in 2019, reflecting a standardized death rate of 21.82 per 100,000 female inhabitants. In addition, in 2017, Poland was fifth in Europe in terms of the deaths number from BC and eighth place because of the mortality rate. The Polish study [5] indicated the constantly growing BC mortality trends since 2010.

The Supreme Audit Office in Poland conducted a number of audits evaluating the implementation of selected tasks related to the cancer control process. A common element of these controls was the assessment of securing the health needs of citizens through the cancer prevention and treatment system functioning in Poland. According to the findings, there has not been sufficient improvement in the key areas determining the effectiveness of oncological treatment (i.e., detection of the disease at the earliest possible stage, providing all patients with access to fast and coordinated diagnostics and application of optimal therapeutic procedures) allowing the achievement of European patient survival rates. The following were identified as the main factors

low public awareness of the causes of carcinogenesis;poor organization of the health care system (lack of basic preventive examinations, deteriorating access to services, no reference centers, an inefficient information flow system that does not guarantee the physician access to information about the patient’s health, lack of treatment coordination);limited and inadequately deployed resources of the health care system, in particular medical staff, as well as specialized equipment [8].

Cancer is a large and growing source of economic burden with above EUR 2.6 billion in associated medical care costs estimated in Poland in 2020 [9]. To compare, in 2009 Poland spent on oncology EUR 1.1 billion, which means over 230% of increase over a decade. The total cost of cancer and its development over time depends on the size of the disease burden—incidence, prevalence, and mortality, as well as technological progress. Population aging and better diagnostic methods are driving the rising number of new cancer patients, which leads to an increase in the health care expenditures for treatment. New therapies can improve patient outcomes, including reduction of mortality in working-age patients which can reduce productivity loss. On the other hand, the newest methods of treatment are expensive and typically require additional health care spending.

Breast cancer is one of the most costly cancers for the health care system. In the USA, the expenditures for BC care in 2020 were equal to USD 29.8 billion, and these costs included national expenditures for medical services and prescription drugs and represented nearly 16% of costs of all cancers [10]. Total health system expenditure for BC in Australia in 2015–2016 was equal to AUD 1056 million, which includes AUD 269 million for the national screening program [11]. In Portugal, in 2014 the direct cost attributed to BC was EUR 146.1 million, about 1.4% of all Portuguese health expenditures [12]. Italy study [13] from 2008–2016 showed that the National Health Service’s average annual expense related to patients with hospital admissions for BC was estimated at EUR 260 million in 2016 (cost of 103,564 hospitalizations in 75,952 patients—average 1.4 hospitalization per patient in 2016). The structure of the BC treatment costs in Romania showed that three-quarters of the cost was spent on cancer drugs and radiotherapy, while 17.8% of the total cost was spent on hospital care [14].

The main objective of this study was to determine the current direct costs of the diagnosis and the treatment of patients with BC in Poland from a public payer perspective based on the data of the National Health Fund (NHF) estimated for the 2017–2019 years. Additionally, indirect costs were presented based on expenditures on sick leaves reported by the Social Insurance Institution. Finally, the identification of the key cost-consuming components with an analysis of cost changes in time was performed.

## 2. Materials and Methods

Cost estimation was prepared using a top-down approach because the aggregate cost data and resource utilization were taken mainly from official databases. The top-down approach is typically based on financial and accounting data, therefore resource utilization is measured retrospectively. This approach is simpler and more transparent. Moreover, it is faster and cheaper than the bottom-up approach, which requires patient-level data [15].

The health care system in Poland has been divided into 4 main sectors of healthcare: primary health care (GPs), outpatient specialist care, inpatient health care—hospitals and the emergency medical care with several special areas (long-term care, rehabilitation facilities). The public health care is free of charge for the patients. National Health Fund is the single national payer for health care services. The outpatient and inpatient care are funded by NHF in form of diagnosis-related groups. NHF is also responsible for the adjustment in prices of the services and all changes are published.

Data from the NHF and Social Insurance Institution databases for the ICD-10 code C50 Malignant neoplasm of breast were collected for the years 2017–2019. For the cost categories BC screening program, chemotherapy, radiotherapy, and outpatient care (which were published without ICD-10 diagnosis codes), data for code C50 were provided by the NHF, upon the authors’ request. The remaining cost data were taken from published NHF databases [16,17,18].

The expenditures for BC in Poland consisted of direct medical care costs and indirect costs. Medical care costs included the following components:BC screening program—the first step was the basic diagnostic in the form of mammography, after which patients with abnormal results were referred for extended diagnostics (enhanced diagnosis stage), which could include clinical examination of the breast, fine-needle or core needle breast biopsy;Oncological package—also called the Rapid oncological therapy package for which the main goal was to efficiently and quickly guide the patient through the next stages of diagnosis and oncological treatment. Initially, the patient was diagnosed, and then the treatment method was determined on the Concilium. The BC Concilium was a type of medical consultation with several clinical specialists: a breast tumor surgery specialist, a clinical oncology specialist in BC, a specialist in x-ray diagnostics or radiology, and a specialist in radiotherapy. During the entire treatment process, the patient was under the constant care of the coordinator. Only costs of basic diagnostic and enhanced diagnosis under this package were included in the calculations. Other costs are taken into account at different stages of treatment, e.g., radiotherapy and outpatient treatment.Surgical treatment—breast-conserving surgery (BCS), mastectomy, and breast reconstruction. The expenses of the indicated types of surgical procedures were estimated based on the Polish Diagnosis-Related Group (DRG) system. DRG related to BC were: J01 Radical breast removal with reconstruction, J02 Complex breast surgeries, J03E Major breast surgeries >65 years of age, J03F Major breast surgeries <65 years of age, J04 Breast reconstruction treatments, and J05 Medium breast surgery. BCS was realized with groups J01, J02, J03E, and J03F; mastectomy with groups J01, J02, J03E, J03F, and J05; and breast reconstruction with group J04.Hospitalization—conservative treatment (DRG: J08. Malignant breast diseases) and mammotomic biopsy (DRG: J10. Mammotomic biopsy);Drug program (also called Therapeutic program)—B.09 Treatment of patients with breast cancer (ICD-10: C50). Drug program costs included costs of drugs (trastuzumab, pertuzumab, lapatinib, palbociclib (refund has started in 2019) and ribociclib (2019)), hospitalization or outpatients visit related to the patient treatment in the drug program, and diagnostic tests which have determined the efficacy and safety of the drugs in the program;Chemotherapy (CHT)—treatment provided to patients with diagnosis ICD-10: C50.X, including the cost of chemotherapeutic agents and the cost of CHT administration (e.g., hospitalization, outpatient consultations);Radiotherapy—costs of teleradiotherapy, brachytherapy, and radioisotope therapy in patients with diagnosis ICD-10: C50.X. This category also included the costs of accommodation, hospitalization, and treatment of adverse events related to radiotherapy in patients with BC;Outpatient care—all outpatient visits and medical services provided to patients with diagnosis ICD-10: C50.X. In particular, these were diagnostic tests on an outpatient basis (e.g., nuclear medicine tests, computed tomography, magnetic resonance imaging), outpatient visits, or outpatient surgical services.

Medications refunded within the drug program and chemotherapy were available to the patient free of charge. All other included medical services were fully financed by NHF.

The indirect costs were social costs resulting from absenteeism caused by sick leaves due to BC (ICD-10 code: C50). Indirect costs were calculated based on data from Statistics Poland [19] usage gross domestic product (GPD), the number of employees and based on the Social Insurance Institution database, the number of sick leave days [20] using the human capital approach. Calculation of productivity lost was conducted following the Polish recommendations [21]. Direct and indirect costs were gathered and calculated in Polish zlotys (PLN). The expenditures were expressed as Euro (EUR) using the mean 2017–2019 exchange rate: 1 EUR = 4.2726 PLN [22]. The price indices in “Health” area were 102.6 at the end of the year in 2018 and 103.7 in 2019 for the same period of the previous year = 100. Inflation was insignificant that it was not decided to take it into consideration. As the costs were taken from the NHF databases, it also included all adjustments in the price of the included services.

## 3. Results

Total expenditures for BC including direct costs (diagnosis and treatment BC) and indirect costs (social cost) amounted to EUR 305,371, EUR 332,998, and EUR 344,649, respectively in 2017, 2018, and 2019. The percentage of the medical care costs in total expenditures was decreasing from 69% (EUR 209,527) in 2017 to 65% in 2019 (EUR 224,284). At the same time, the shares of social costs increased year to year. (Table 1, Figure 1).

Our findings showed that total expenditure for medical care of BC was 9% and 7% higher, respectively, in 2018 and 2019 in comparison with costs in 2017. However, total healthcare costs in 2019 were EUR 4114 lower than in 2018 because of the reduction in expenditure on the therapeutic program (EUR 13,526.7), despite the observed increase in all remaining resources. (Table 1, Figure 2). 

The highest expense was spent on the drug program (nearly 50% of total direct costs), where drug costs represent more than 92% of annual program reimbursement costs. By comparing the results obtained in 2018 and 2019 for drug program B.09, it should be pointed out that in this period the number of patients increased by 10% while the costs were reduced by 12%. The reduction probably resulted from the reimbursement of seven generic drugs in 2019. At the end of this year, a total of 16 drugs were on the official drug list and only 7 (including 2 generics) and 5 (all original drugs) at the end of 2018 and 2017, respectively [23,24].

Significant cost increases have been observed for surgical treatment. BCS costs in 2018–2019 amounted to over EUR 15,000 (an increase of 37% compared to 2017). At the same time, only 233 hospitalizations more than in 2017 were observed in 2019, but in 2018 there were almost 2.5 thousand more inpatient treatments. It should be noted that the average cost of a single hospitalization increased year by year, comparing 2017 to 2019. The cost of one BCS increased from EUR 890 to EUR 1200 (35% increase); cost of mastectomy, from EUR 1100 to EUR 1650 (50% increase); and cost of breast reconstruction, from EUR 820 to EUR 1120 (37% increase). (Table 1, Figure 3). It resulted mainly from the higher funding of BC DRGs changed by NHF.

Total costs associated with hospitalizations due to conservative treatment and mammotomic biopsy were also increasing. The annual number of conservative treatments increased from year to year by approximately 200 hospitalizations, while the number of biopsies remained at around 6000 services/year (Table 1).

The costs of breast cancer CHT in 2018 and 2019 remained at a similar level and amounted to EUR 10,106–10,340, respectively (an increase of 16.3–19.0% compared to 2017). Annually, approximately eight hundred thousand outpatient visits and diagnostic tests were reported in patients with BC diagnosis. Outpatient care cost NHF budget of EUR 18,306, EUR 19,791, and EUR 21,078 in 2017, 2018, and 2019, respectively (Table 1).

For the screening program, the cost of basic diagnostics represents more than 94% of total expenditures each year. Despite the slight increase in expenditure on mammography in 2019 (4.3% and 5.5% in comparison to 2017 and 2018, respectively) the number of the diagnostic tests remained at the same level: slightly over 1 million per year. (Table 1, Figure 4).

The costs of enhanced diagnosis account for 84% (in 2017), 77% (in 2018), and 69% (in 2019) of total oncological package (diagnostic) expenditures. During the observation period, a steady increase in the annual number of basic and extended diagnostic tests was observed. This corresponded to an increase in expenditure on basic diagnostics, but a decrease in costs for extended diagnostics in 2019 (less by 7% and 10% compared to costs in 2017 and 2018, respectively) (Table 1, Figure 4).

For the radiotherapy category, teleradiotherapy cost represents more than 91% of total expenditures each year, and approximately 5% of teleradiotherapy costs were for palliative treatment. An increase in costs in time was observed: from EUR 20,017 in 2017 to EUR 26,229 in 2019, wherein the average cost for one service changed slightly (2% increase, 2019 vs. 2017).

In our analysis, we also included the costs of absenteeism caused by the BC (the social cost of the disease) (Table 1). In 2019 the BC generated nearly EUR 345 million of social costs. In 2019, compared to 2017, there was an increase in the value of productivity loss by 26%. In 2019, more than 30% of the total expenditures for BC management in Poland were generated by indirect costs (costs of absenteeism) (Figure 1). It was also observed that the number of sick leave days, as well as the number of medical certificates, increased from year to year (Figure 5). The number of medical certificates due to BC in 2019 in Poland was equal to 57,179, which was an increase of 18.1% from 2017. The average length of sick leave in 2019 was equal to 22.2 days.

## 4. Discussion

Breast cancer represents a significant source of disease burden in Poland, both clinically and economically. According to data provided by Statistics Poland, the public expenditures on health care were equal to EUR 24,836 million in 2019 [25]. The total cost for medical health care for BC in Poland in 2019 was estimated to be EUR 224.3 million, which means that in 2019 the medical costs of BC represented 0.9% of total health care expenditures. To compare, in the Portuguese study from 2014 1.4% of total health expenditures were spent on BC treatment [12]. In 2019 total health care expenditures per capita in Poland were 70% of those in Portugal. Therefore, if the level of expenditure on BC treatment in Portugal would remain at the 2014 level, then the costs of BC treatment would be more than twice as high as in Poland.

For both Italian [13] and Romanian [14] conditions, hospital care costs accounted for 17–18% of the total medical costs of breast cancer treatment. Similarly in Poland, hospitalization costs were estimated at 12.5–18% in 2017–2019.

In 2019, 10% of BC medical costs were determined by the screening program (Figure 4). When comparing our results to those of AIHW from 2016–2017, it must be pointed out that in Australia a screening program represented more than 25% of BC health care expenditures—approximately 2.5 times more than in Poland [11]. In addition, the World Health Organization (WHO) determines that the effectiveness of the BC screening program depends on its high quality and ability to offer to screen at least 70% of the eligible population [26]. In Poland, in the analyzed period, the percentage of the population covered by screening did not exceed 40%. During the two years of the COVID-19 pandemic, this percentage decreased by another 5 percentage points [27]. Therefore, it seems that the expenditure on promotion and implementation of the program should be increased almost twice.

Costs per disease stage could not be presented because the cost data gathered by the NHF does not allow such costs to be distinguished. The Polish public payer only requires ICD-10 classification codes to be used for reporting in its settlements.

The B.09 drug program was the most cost-consuming element of direct cost components. This drug program had also the largest number of patients (almost 25% of patients) and the biggest cost (~19% of total expenditures) of all oncologic drug programs in Poland (29 drug programs at the end of 2019). However, when comparing the average cost per patient in all of those drug programs, B.09 ranked only 16th (second half) in the most expensive per-patient programs [17].

Considering the total NHF costs for oncological treatment in Poland [9], 10.2% was allocated to the BC treatment in 2017–2018, while in 2019, one percentage less (9.2%). According to the American study, BC costs represented 16% of expenditures for all cancers [10], and that is over 50% more than in Poland.

The BC indirect costs included only the costs of productivity lost due to sick leaves. A comprehensive analysis of the indirect costs attributable to breast cancer in Poland was conducted by the authors of the publication Łyszczarz 2017 [28], where social costs included productivity losses and the public finance burden attributable to breast cancer in Poland. The social costs included absenteeism of the sick, presenteeism of the sick, caregivers’ absenteeism and presenteeism, premature mortality, and disability. Absenteeism of the sick, which in addition to the costs of sick leaves also considered the cost of rehabilitation benefits, accounted for only 18% of total social costs. Of the six indirect costs categories, losses due to disability, premature mortality, and caregivers’ presenteeism caused the highest economic burden (over 70% in total). If the costs of absenteeism constitute a maximum of 20% of indirect costs, the estimated costs of productivity lost in 2019 will be over EUR 585 million. It means that the total indirect costs may exceed by more than 2.5 times the direct costs of treatment BC.

## 5. Conclusions

Our results from 2017–2019 demonstrated that total expenditures for BC in Poland increased from year to year. In 2019, the cost of BC was equal to EUR 344.6 million, which includes EUR 224.3 million in medical care expenditures and EUR 120.4 million of social costs (absenteeism). In each year, nearly 50% of total costs were represented by drug program B.09. It is worth noting that only for this treatment method, the important decrease—8% of total expenditures from 2017 to 2019—was observed, mainly because of the introduction the generic drugs. A significant increase in this period was found for surgical treatment (~4%) and radiotherapy (~2%).

## Figures and Tables

**Figure 1 ijerph-19-16384-f001:**
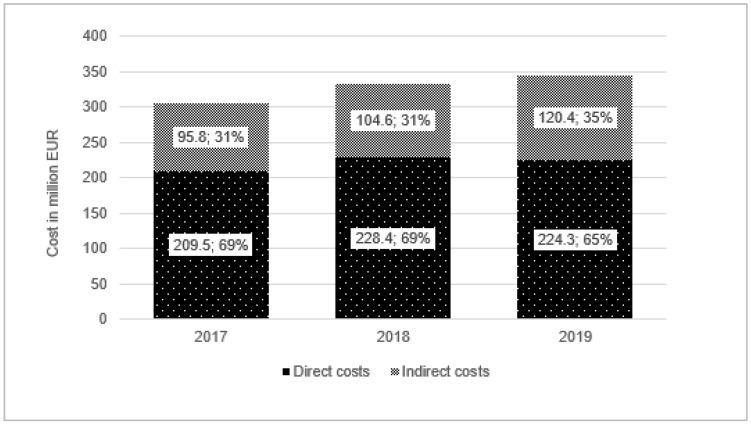
Total expenditures for breast cancer treatment in Poland.

**Figure 2 ijerph-19-16384-f002:**
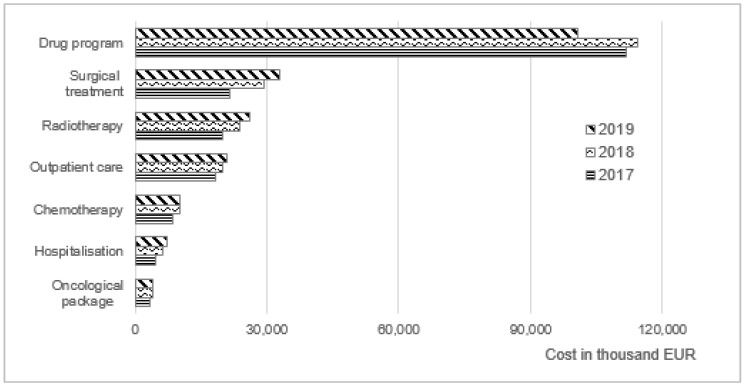
The direct costs of medical care of breast cancer in Poland.

**Figure 3 ijerph-19-16384-f003:**
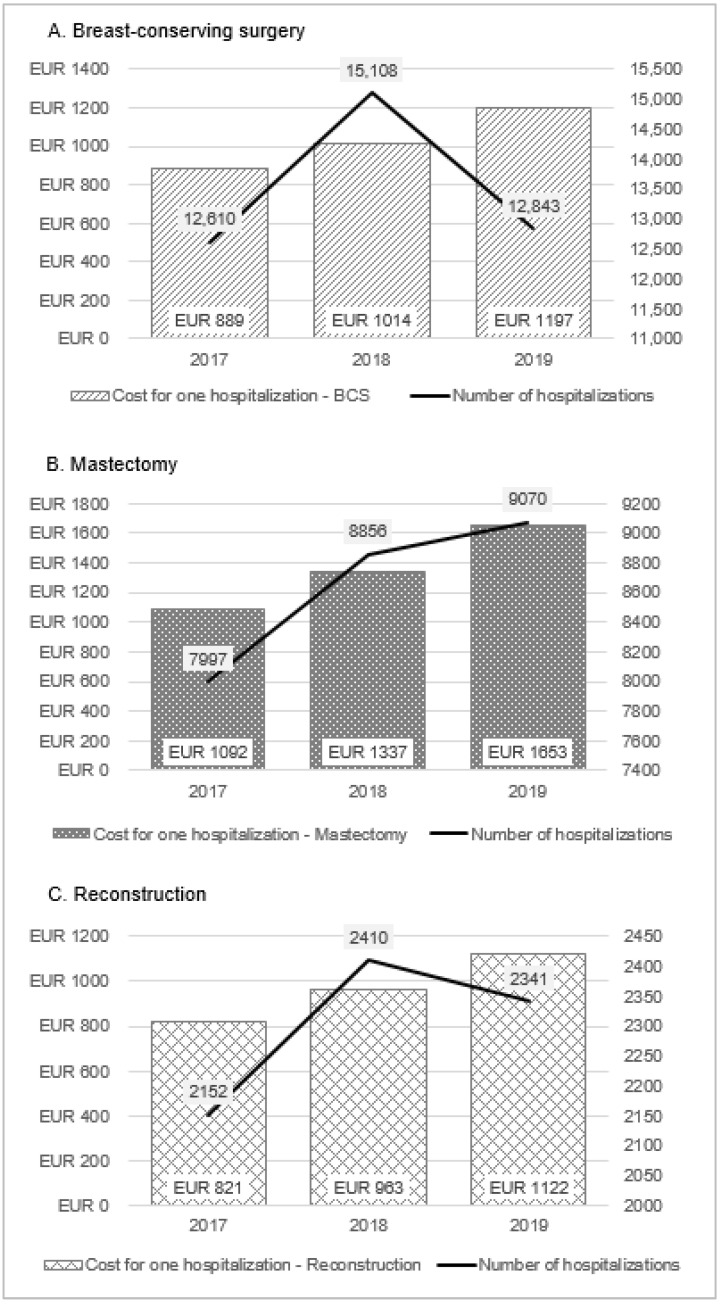
Costs of one hospitalization of surgical treatments.

**Figure 4 ijerph-19-16384-f004:**
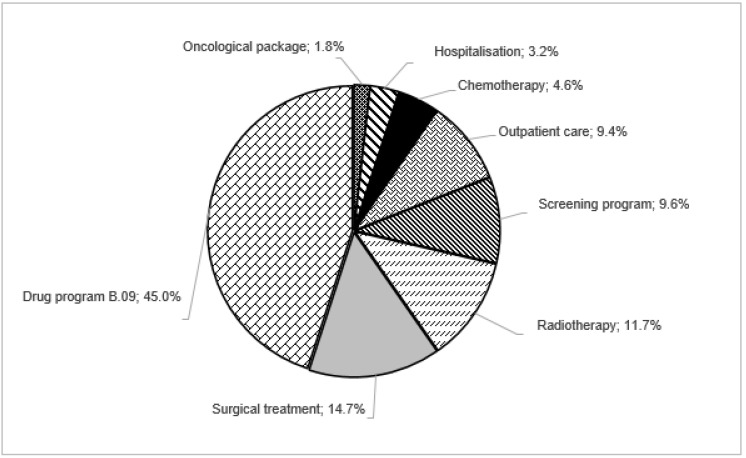
Distribution of direct costs components (2019).

**Figure 5 ijerph-19-16384-f005:**
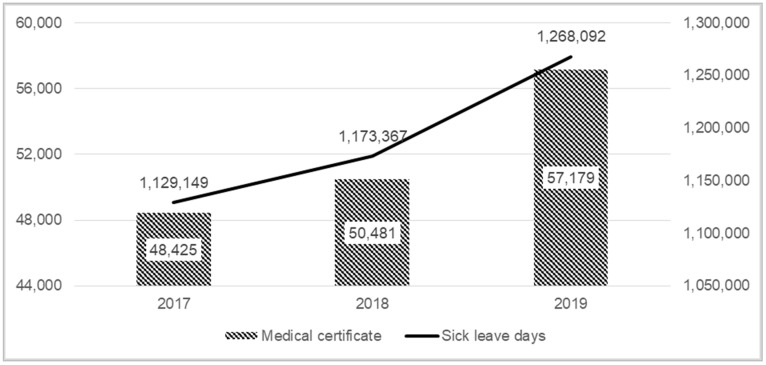
Sick leave days and medical certificate.

**Table 1 ijerph-19-16384-t001:** The cost statement for breast cancer in Poland.

Treatment Method	Resource Utilization, Healthcare Services Number			Expenditures of the Public Payer (in Thousand EUR)		
	2017	2018	2019	2017	2018	2019
	**Direct costs**					
BC screening program ^1^				20,654.0	20,280.4	21,483.6
Basic diagnostic stage: screening mammography	1,048,151	1,017,165	1,043,743	19,522.0	19,221.1	20,357.1
Enhanced diagnosis stage	34,269	31,894	33,241	1132.0	1059.3	1126.5
Oncological package (only diagnostic) ^2^				3538.5	4013.1	4010.9
Basic diagnostic	6374	10,828	14,589	554.2	919.6	1238.0
Enhanced diagnostics	14,682	15,417	15,569	2984.3	3093.5	2773.0
Surgical treatment ^2^				21,713.2	29,482.6	32,994.8
Breast-conserving surgery (BCS)	12,610	15,108	12,843	11,211.9	15,324.7	15,374.1
Mastectomy	7997	8856	9070	8733.9	11,837.3	14,993.4
Breast reconstruction ^3^	2152	2410	2341	1767.4	2320.5	2627.3
Hospitalization ^2^				4761.3	6453.7	7318.9
Conservative treatment	7503	7747	7990	2499.0	3011.6	4185.5
Mammotomic biopsy ^4^	6257	6825	5838	2262.4	3442.1	3133.5
Drug program B.09 ^2^				111,851.5	114,357.8	100,831.1
Drugs (number of patients)	6883	7341	8077	104,529.0	106,551.8	93,148.0
Hospitalization and outpatients visit	60,747	63,542	62,301	5979.2	6095.9	5803.5
Diagnostics tests	2933	3215	3250	1343.3	1710.1	1879.6
Chemotherapy ^1,5^	na	na	na	8685.8	10,105.7	10,337.7
Radiotherapy ^1,5^				20,016.9	23,914.3	
Teleradiotherapy	33,790	39,135	43,795	18,272.7	22,136.3	24,153.9
Brachytherapy and radioisotope therapy	913	941	1067	1744.1	1778.0	2075.3
Outpatient care ^1,5^	756,992	772,353	806,582	18,306.1	19,790.8	21,078.0
Total medical care cost				209,527.3	228,398.4	224,284.3
Year-to-year change				-	9%	−2%
	**Indirect costs**					
Number of sick leave days/Number of medical certificate	1,129,149/ 48,425	1,173,367/50,481	1,268,092/57,179	-		
Value of productivity loss due to BC (social cost)				95,844.1	104,599.8	120,364.3
Year-to-year change				-	9%	15%
Total expenditures for BC				305,371.4	332,998.2	344,648.5
Percentage of medical care cost in total expenditures				69%	69%	65%

^1^ EUR = 4.2726 PLN, National Bank of Poland (NBP) average exchange rate: 2017–20191 data provided by the National Health Fund (NHF) in Poland; ^2^ data published by the NFH in Poland; ^3^ breast reconstruction can be also performed as part of DRG J01na–not available. ^4^ for the year 2017 data from https://statystyki.nfz.gov.pl/Benefits/1a (accessed on 17 August 2020), for years 2018–2019 data from www.ezdrowie.gov.pl (accessed on 21 July 2020); ^5^ data for patients with diagnosis ICD-10: C50.X.

## Data Availability

The data analyzed during the current study are available from the authors.

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
