# Peer review of "The Direct and Indirect Costs of Breast Cancer in Poland: Estimates for 2017–2019"

_ijerph, 2022, doi:10.3390/ijerph192416384_

Round 1
Reviewer 1 Report
The manuscript presents a prevalence-based cost study from Poland for the years 2017-2019 and shows a linear increase in costs in this period. The study shows the cost increase according to cost categories, which is a valuable information for policymakers and useful for future health economic studies. However, the manuscript lacks important methodological detail that compromises the interpretation of results.
Introduction
The foreigner reader would benefit from an overview of the health care system in Poland, so can understand the scope of the National Health Fund.
Page 2, line 71-73: It is not clear what the authors wanted to convey I these lines, reformulation is strongly recommended.
Page 2, line 93-105: This is an interesting overview of costs of breast cancer in other countries, but the comparison is already well-explored in the discussion.
Methods
When costs from different year are compared, are cost data from different time periods adjusted for inflation with a specified inflation rate? Was discounting considered?
Page 4, line 176-178: was productivity loss valued differently for subpopulations, e.g. employed/ retired individuals? Which analytic method was used: human capital approach, friction cost or other?
Results:
Page 9, line 227-228: The authors mention that “It should be noted that the average cost of a single hospitalization increased year by year”. Is this change relative to the inflation or where these price adjustments in the valuation of these procedures? Additional description on the methods part would help clarify these statements.
Did the authors consider assessing costs per disease stage? If not, these should be discussed as a limitation.
Author Response
Nov 26, 2022
Dear Reviewer 1,
Please find enclosed a revised version of our manuscript entitled “The direct and indirect costs of breast cancer in Poland estimates for 2017-2019 years” (Sub Id ijerph-1994880) which I am re-submitting for consideration for publication in the International Journal of Environmental Research and Public Health.
We sincerely appreciated the positive feedback and constructive criticisms of the Reviewers and have revised our paper accordingly. Please find a point-by-point response to the reviewer’s comments below.
Please feel free to contact me with any questions concerning this manuscript. We look forward to your response.
The manuscript presents a prevalence-based cost study from Poland for the years 2017-2019 and shows a linear increase in costs in this period. The study shows the cost increase according to cost categories, which is a valuable information for policymakers and useful for future health economic studies. However, the manuscript lacks important methodological detail that compromises the interpretation of results.
@ Thank you for this feedback – the manuscript was revised, and we do hope all your comments were responded correctly.
Introduction. The foreigner reader would benefit from an overview of the health care system in Poland, so can understand the scope of the National Health Fund.
@Thank you for this comment – the manuscript was supplemented with a synthetic overview of the Polish health care system in lines 127-133 (in the settings section).
Introduction. Page 2, line 71-73: It is not clear what the authors wanted to convey I these lines, reformulation is strongly recommended.
@ Thank you for this remark – the relevant part of the text was expanded and completed in lines 71-82.
Page 2, line 93-105: This is an interesting overview of costs of breast cancer in other countries, but the comparison is already well-explored in the discussion.
@ Thank you for this comment. In the introduction part we presented the review of the costs of breast cancer from the other countries. It served as a justification for the aim of our study: that BC is a health problem generating a huge economic and financial burden for the public payer and society. Additionally, in the discussion we focused on comparing the results of foreign costs studies with our results. We pointed out the differences and possible explanations of such situation. I you find our explanation convincing, we would like not to change this part .
Methods. When costs from different year are compared, are cost data from different time periods adjusted for inflation with a specified inflation rate? Was discounting considered?
@ Thank you for this valuable comment. In our paper, we presented the real costs of treatment from the chosen years. The inflation in years 2018-2019 was insignificant in that time (2,6-3,7%), that we decided to not take it into consideration. The additional clarification was included in lines 195-198.
Methods. Page 4, line 176-178: was productivity loss valued differently for subpopulations, e.g. employed/ retired individuals? Which analytic method was used: human capital approach, friction cost or other?
@ Thank you for this comment. The indirect costs were calculated only as the absenteeism caused by sick leaves (lines 188-189). We used human capital approach, and the manuscript was supplemented with this clarification (line 192).
Results: Page 9, line 227-228: The authors mention that “It should be noted that the average cost of a single hospitalization increased year by year”. Is this change relative to the inflation or where these price adjustments in the valuation of these procedures? Additional description on the methods part would help clarify these statements.
@ Thank you very much for this important remark. This increase in the average cost of a single hospitalization resulted mostly from the small increase in prices during the period under analysis – the funding of DRGs for BC treatment was increased by NHF. We added additional explanation in the settings section (lines 198-199) and the clarification in the line 247.
Did the authors consider assessing costs per disease stage? If not, these should be discussed as a limitation
@ Thank you for this valuable comment. We added in the limitation section the justification why it was not possible to perform in this study (lines 314-316).

Reviewer 2 Report
This is a very useful and interesting paper that reads very well.
I missed knowing if Oncotype is approved/performed in Poland based on this paper as it is not mentioned; the cost of the test is high but the prevention of chemotherapy administration may impact final costs per patient and I believe this should be mentioned (if test is performed vs not and the cost).
Another thing worth exploring is the cost and loss of productivity of treatment of primary disease vs metastatic disease as even if more costly (prevention and early phase treatment), this can overcome costs and loss of productivity of metastatic disease to justify government investments.
Figure 4 subtitle seems to be incorrect as it does not match the content in the figure.
Otherwise, no additional considerations and I very much enjoyed reading the manuscript.
Author Response
Nov 26, 2022
Dear Reviewer 2,
Please find enclosed a revised version of our manuscript entitled “The direct and indirect costs of breast cancer in Poland estimates for 2017-2019 years” (Sub Id ijerph-1994880) which I am re-submitting for consideration for publication in the International Journal of Environmental Research and Public Health.
We sincerely appreciated the positive feedback and constructive criticisms of the Reviewers and have revised our paper accordingly. Please find a point-by-point response to the reviewer’s comments below.
Please feel free to contact me with any questions concerning this manuscript. We look forward to your response.
This is a very useful and interesting paper that reads very well.
@ Thank you for the positive feedback on our study.
I missed knowing if Oncotype is approved/performed in Poland based on this paper as it is not mentioned; the cost of the test is high, but the prevention of chemotherapy administration may impact final costs per patient and I believe this should be mentioned (if test is performed vs not and the cost).
@ Thank you for this remark. This test was assessed by Polish HTA Agency in July of 2020, and it was not approved. Therefore, it is not funded by the Polish public payer and when such technology is not covered by NHF, it is usually not used in the standard clinical practice. Our analysis was focused on the BC diagnosis and treatment costs used in Poland in the period of 2017-2019, and Oncotype in that time was not mentioned in the Polish clinical practice. This is why we decide not to include it into our study.
Another thing worth exploring is the cost and loss of productivity of treatment of primary disease vs metastatic disease as even if more costly (prevention and early phase treatment), this can overcome costs and loss of productivity of metastatic disease to justify government investments.
@ Thank you for this comment. We agree that such approach to indirect costs of breast cancer would be interesting and informative for the readers. Unfortunately, the Polish data of absenteeism are only available for the ICD code, and it is not possible to calculate it for the disease stage.
Figure 4 subtitle seems to be incorrect as it does not match the content in the figure.
@ Subtitles were updated accordingly.
Otherwise, no additional considerations and I very much enjoyed reading the manuscript.
@ Thank you once again – your opinion is of great importance to us.

Round 2
Reviewer 1 Report
Thank you for the responses, the comments were satisfactorily aswered.